

# Italian adaptation of the Multidimensional Iowa Suggestibility Scale (MISS)

Marco Tullio Liuzza[1,*], Eva Tolomeo[2,*], Giuseppe Occhiuto[1], Martina Cilurzo[1], Iolanda Martino[1] and Antonio Cerasa[3,4,5]

[1] Department of Medical and Surgical Sciences, "Magna Graecia" University of Catanzaro, Catanzaro, Calabria, Italy
[2] Department of Health Sciences, "Magna Graecia" University of Catanzaro, Catanzaro, Calabria, Italy
[3] Institute for biomedical research and innovation, National Research Council, Messina, Sicily, Italy
[4] S. Anna Institute, Crotone, Calabria, Italy
[5] Department of Pharmacy, Health Science and Nutrition, University of Calabria, Arcavacata di Rende, Calabria, Italy
[*] These authors contributed equally to this work.

Corresponding authors
Marco Tullio Liuzza, liuzza@unicz.it
Eva Tolomeo, eva.tolomeo@unicz.it

## ABSTRACT

**Background**. Suggestibility is a personality trait that reflects a general tendency to accept messages. The Multidimensional Iowa Suggestibility Scale (MISS) is a self-report scale developed to measure the degree of individuals' perceptions of their suggestibility. This study aimed to adapt the MISS in an Italian sample.

**Methods**. We conducted two studies. In the first study, 345 subjects (270 females (78%), mean age = 36.21 years ± 14.06 SD) completed the translated Italian version of the MISS, composed of five subscales (consumer suggestibility; persuadability; sensation contagion; physiological reactivity; peer conformity). We investigated the structural validity of the scale through confirmatory factor analysis (CFA) testing four measurement models (unidimensional, four-factor, hierarchical four factors, and bifactor) and explored reliability in terms of internal consistency through the McDonald's omega. In the second study, we cross-validated the MISS on a new independent sample. We enrolled 277 participants (196 females (71%), mean age 30.56, SD = 12.58) who underwent the new version of the scale. We performed factor analyses to test structural validity and compared four measurement models. Then, we investigated reliability and conducted a latent variable analysis to explore divergent validity.

**Results**. The CFA in the first study revealed a bifactor solution of the MISS. This structure was interpretable and provided an adequate fit for the data. The final version of the scale was reduced to forty-six items with globally good indices of adaptation. The scale also demonstrated acceptable reliability in terms of internal consistency through the McDonald's Hierarchical Omega. In the second study, we found that the bifactor structure was confirmed. Factor loadings inspection revealed that there was no justification to report only the separate scores for the subscales. We also found that the scale showed good internal consistency, but mixed evidence for divergent validity.

**Conclusions**. In the end, the Italian version of the MISS demonstrated good psychometric properties which will be discussed in detail below.

# INTRODUCTION

Suggestibility is a personality trait that reflects a general tendency of individuals to accept and internalize messages (*Kotov, Bellman & Watson, 2004*). Suggestion refers to a process of communication in which one or more persons produce a change in one or more individuals' opinions, attitudes, behaviors, and judgments, without the involvement of rational thinking (*Eysenck, Arnold & Meili, 1975*), to the extent that the response to suggestion is usually considered unreasoned and uncritical (*Coffin, 1941*). Research on suggestion and suggestibility originates from the first studies on hypnosis in the nineteenth century. However, it is important to distinguish hypnotic suggestibility—responsiveness to suggestions given after the induction of hypnosis—from nonhypnotic suggestibility—in which the suggestions are given outside hypnosis and are aimed at convincing people that the world is not the same as it actually is (*Braffman & Kirsch, 1999*). With the passing of time, numerous authors have expanded the field of research and, especially sociologists and social psychologists, used the term suggestion to explain some social implications and clarify how propaganda has some influence on the acquisition or change of attitudes (*Gudjonsson, 1987*).

To date, a deep comprehension of suggestibility is lacking and there is little agreement about its dimensionality. Numerous authors (*Eysenck & Furneaux, 1945*; *Grimes, 1948*; *Benton & Bandura, 1953*; *Hammer, Evans & Bartlett, 1963*) have conducted different studies using factor analysis methodologies, but no common structure emerged from them and the question of whether suggestibility is a multifactored construct or not is still unanswered. In a recent study, *Polczyk (2016)* followed a factor analytic approach and found that suggestibility involves two uncorrelated factors, providing new evidence for the existence of direct suggestibility—when the influence is overt and the subject is aware that his susceptibility is being measured—and indirect suggestibility—when the influence is hidden. This is a result that other studies failed to find, probably due to methodological issues. However, as *Oakley et al. (2021)* highlighted, it is reasonable to believe that suggestibility covers a broader domain of traits, of which direct verbal suggestibility constitutes only one of them, while placebo suggestibility and interrogative suggestibility others, as increasing evidence suggests.

A crucial topic is that there is a huge variability among all these studies—in terms of design, sample, and types of measures—that does not allow a confident comparison nor a replication, with the result that all conclusions are uncertain (*Tasso et al., 2020*). There is a shared need to overcome these limitations, understand the real nature of suggestibility, and improve new standardized measures, with a notable benefit in many areas. This is true, especially for clinical practice, where the placebo effect—a phenomenon in which the administration of a pharmacologically inert substance or therapy produces beneficial clinical outcomes—plays a pivotal role, and a deep comprehension of its mechanisms has

important implications. It would allow for improving strategies to control placebo responses minimizing their effect in randomized clinical trials, helping in the interpretation of study results, and maximizing it to improve treatment benefits (*Enck et al., 2013*). We know that placebo responses vary substantially between, and even within, individuals, and this may depend on many factors: cognitive factors, such as expectations and anxiety (*Linde et al., 2007*; *Flaten et al., 2011*), conditioning, prior learning strategies (*Benedetti et al., 2016*) and individual differences. An interesting study by *Parsons et al. (2021)* investigated whether direct verbal suggestibility is a predictor of individual differences in placebo hypoalgesia responsiveness, and found that it depends on the use of conditioning. From their study emerged that suggestibility was significantly correlated with placebo responsiveness when the manipulation only included suggestion and not conditioning.

A further difficulty is that the suggestive process is more complex than it could seem, since it consists of different stages, and involves diverse elements. As *Gheorghiu (1972)* remarks, the process starts with a suggestive stimulus, it continues with suggestibility—the predisposition of an individual to be influenced—and, if present, leads to a suggestible reaction. Except for auto-suggestion, the stimulus derives from interpersonal interactions, which has some important implications. The first is the importance of context which can accentuate the nature of interaction; the second concerns a wide range of motives able to facilitate the suggestive influence, such as expectations, wishful thinking, and interpersonal trust (*Gheorghiu, 1972*; as cited in *Gudjonsson, 1987*).

A crucial question arises "about the relationship between suggestibility and private acceptance of the suggestions" (*Gudjonsson, 1987*). Individuals may accept the suggestions knowing that they are false, implying that they are behaving as others expect them to. In this case, it would be more appropriate to talk about compliance, since there is only a behavioral change without the internalization of a message (*Kotov, Bellman & Watson, 2004*). On the other hand, individuals may accept the suggestions when they privately believe them to be true and this implies a non-conscious volitional decision. The problem is that it is difficult to understand which is the case since there could not be observable differences in their behavior (*Gudjonsson, 1987*). For this reason, adequate instruments are required. In fact, over time, numerous attempts to develop instruments to measure suggestibility have been made.

At first, they were limited to reproducing simple motor and sensory reactions, then more complex stimuli to reach changes in opinions and attitudes. In this regard, the few developed scales investigating attitudes mainly focus on specific domains of suggestibility. For example, we can mention the "Gudjonsson Suggestibility Scale" created by G. Gudjonsson which investigates interrogative suggestibility, especially in the forensic field, and the "Barber Suggestibility Scale" developed by T. X. Barber, which focuses on the hypnotic field. It is also worth mentioning the "Harvard Group Scale of Hypnotic Susceptibility, Form A" (HGSHS:A; *Shor & Orne, 1962*). This scale started as a group version of Weitzenhoffer and Hilgard's "Stanford Hypnotic Susceptibility Scale: Form A" (SHSS:A; *Weitzenhoffer et al., 1959*), and nowadays is the most widely used instrument in hypnosis research (*Angelini, Kumar & Chandler, 1999*), also in Italy (*De Pascalis, Russo & Marucci, 2000*).

From a historical point of view, these scales were based on the Friedlander-Sarbin Scale (*Friedlander & Sarbin, 1938*), which was the first standardized scale used to measure hypnotic susceptibility. Over the years numerous changes have occurred. Initially, the SHSS:A required a large amount of time, as it had to be administered individually and was therefore impractical. In contrast, the HGSS:A was used as a preliminary screening device to provide an initial estimate of hypnotic responsiveness (*Kallio, 2021*). However, both scales have undergone further modification. Despite its popularity, the HGSS:A has received several criticisms. Most of them concern the high presence of false-positive responses in some subscales (*e.g.*, posthypnotic motor movement), or the problematic interpretation of behavioral components. Furthermore, some scales conflate non-hypnotic and hypnotic suggestibility through the inclusion of non-hypnotic suggestions (*Acunzo & Terhune, 2021*). However, whereas the SHSS:A solely focuses on hypnotic suggestibility, recent studies suggest that hypnotic suggestibility is part of a wider psychological trait of direct verbal suggestibility (DVS), which can be investigated even without hypnotic induction (*Oakley et al., 2021*). With this premise, *Oakley et al. (2021)* suggest that by making minor changes, the Harvard Group Scale could be used to measure DVS, but not everyone completely agreed with this solution. In light of this consideration, *Kallio (2021)* suggested that, rather than revising the currently existing scales, a possible solution could be to completely revise the nature of the concept of "suggestibility" itself.

On the other hand, a wider assessment of suggestibility can be made using the Multidimensional Iowa Suggestibility Scale (MISS). The MISS is a self-report scale about individuals' perceptions of their suggestibility based on various statements, devised by *Kotov, Bellman & Watson (2004)* in the United States. The underlying hypothesis is that suggestibility is a personality trait halfway between voluntary and automatic information processing. For the authors, it would also consist of a tendency to accept without particular pressure messages coming from ourselves, from others, or any type of media. After various revisions and analyses, the MISS is presented in its final version consisting of sixty-four items, divided into five subscales: consumer suggestibility, persuadability, sensation contagion, physiological reactivity, and conformism. There are also two companion scales—psychosomatic control, and stubborn opinionatedness—which bring the items to a total of ninety-five. In their validation study, *Kotov, Bellman & Watson (2004)* showed that the MISS displayed reliabilities of $\alpha$s >.80 for all the final scales, except persuadability and mental control ($\alpha = .79$ each). They also provided some evidence for its construct validity, investigating correlations between MISS and self-report measures of Big Five, dependency, absorption, dissociation, obsessive-checking, self-concept clarity, and self-monitoring. Results indicated that the suggestibility scales are distinct from the Big Five traits ($r < .32$), and showed associations with other constructs, but they were not redundant with any of them ($r < .48$). However, the scale has not been translated and adapted in Italian.

To fill this gap, this study aims to adapt the Italian version of the MISS to a large sample of Italian healthy individuals, exploring its reliability, validity, and factor structure. To this purpose, in the first study, we translated and administered the Italian adaptation of the MISS to an Italian sample, tested for its structural validity, and refined the scale. In a

second study, we cross-validated the adapted scale to another sample. In the end, the final version of the MISS showed good psychometric properties.

## MATERIALS & METHODS

### Study 1

#### Participants

The Italian version of the MISS was administered to 345 participants (270 females, 78%), aged 19 to 75 years (mean age = 36.21 years ± 14.06 SD). Nine participants (3%) received a secondary school education, 154 (45%) a high school diploma or equivalent, 17 (5%) a bachelor's degree, 165 (48%) a master's degree and/or a doctorate. Exclusion criteria were: illiteracy and a history of neurological and/or psychiatric disorders. We recruited participants from universities, community recreational centers, and hospital personnel through local advertisements.

#### Procedure

This study was approved by the Ethics Committee of the Calabria Region (Registration protocol n. 285 del 17 Ottobre 2019) and the procedure was carried out according to the Declaration of Helsinki. All participants were provided with a clear explanation of the aim of the examination and gave their written informed consent to participate in the study.

#### The Italian version of the multidimensional iowa suggestibility scale

The study consists of a four-stage process of translation: (1) forward translation (2) back translation (3) modification of expert reviews and (4) adaptation. Initially, the scale was translated into Italian. Two independent literal translations by two bilingual Italian native speakers, one with a psychological background and a naïve translator without a psychological background, were carried out. Content, meaning, clarity of expression, and comparability to the original item were verified. Back translation to the original language was produced by two native-English-speaking translators, blind to the original questionnaire and without psychological background. The last author addressed any differences between the two English language documents, comparing the back translation to the original. The questions that showed discrepancies could be rephrased and the process could repeat several times. Finally, the goal was to ensure that the translation captures the closest possible meaning of the original item, and the questionnaire was tested for cross-language equivalence.

Participants completed an Italian version of the MISS (*Kotov, Bellman & Watson, 2004*), a self-report scale consisting of sixty-four items. The authors have permission to use this instrument from the copyright holders. Each item consisted of a statement on which the participants had to express their degree of agreement on a Likert scale from 1 (*Strongly disagree*) to 5 (*Strongly agree*). The MISS includes five suggestibility subscales: Consumer Suggestibility; Persuadability; Sensation Contagion; Physiological Reactivity; and Peer Conformity. Eleven items belonged to the subscale "Consumer" (*e.g.*, "I often get information about products from commercials"), fourteen to the subscale "Persuadability" (*e.g.*, "I am easily influenced by other people's opinions"), twelve to the subscale "Sensation

Contagion" (*e.g.*, "When someone describes an experience, I sometimes feel as if I am having it"), thirteen to the subscale "Physiological Reactivity" (*e.g.*, "After seeing a scary movie I feel jumpy for a while"), fourteen to the subscale "Peer Conformity" (*e.g.*, "I share many of my friends' opinions").

## Data analysis

**Descriptive analysis.** We explored the main descriptive statistics (mean, standard deviation, median, range, kurtosis, and skewness) and performed the test of *Mardia (1970)* to verify the assumption of multivariate normality. *P*-values greater than .05 indicate the assumption is met.

**Factorial structure.** To test the hypothesized relationship between latent and observed variables we tested four models and performed, for each model, a confirmatory factor analysis (CFA) for ordinal data through a diagonally weighted least square estimator (DWLS), using the *lavaan library* (*Rosseel, 2012*) on the R statistical programming environment (*R Core Team, 2022*). We tested four measurement models: (1) one unidimensional (for comparison only, since we already knew from the original validation study that the MISS was multidimensional), where all items load onto the suggestibility factor, (2) another with four factors, represented by the four subscales of the MISS, (3) a hierarchical second-order latent variable model (SOLV) with the same four factors as first-order factors loading on a higher-order factor (suggestibility), and (4) a bifactor model, with a general factor (in our case, suggestibility) and four specific grouping factors (the subscales of the MISS). The bifactor model is characterized by a general factor that is reflected by all the observable variables and some grouping factors which are reflected by specific sub-groups of the same observed variables. In particular, the bifactor model is a precious tool to assess the underlying dimensionality of a test, investigate how variance is partitioned when it is supposed to derive both from a general factor and specific grouping factors, and study the adequacy of a total score (*Reise, Moore & Haviland, 2010*; *Reise, 2012*; *Reise, Bonifay & Haviland, 2013*).

**Model estimation and evaluation.** Comparative Fit Index (CFI), Tucker-Lewis Index (TLI), Root Mean Square Error of Approximation (RMSEA), Standardized Root Mean Squared Residual (SRMSR), and the ratio of chi-squared to degrees of freedom ($\chi 2/df$) were used as indices to analyze the goodness of fit of the models. The CFI and TLI values >.90 were considered good, while the absolute fit index values of RMSEA and SRMR values <.10 were considered good (*Hu & Bentler, 1999*), and values of $\chi 2/df$ <3 were considered acceptable. The best model was chosen using (i) an evaluation of the fit indices; and (ii) the $\chi 2$ difference test between models, with a *p*-value lower than 0.05 indicating a difference between the two compared models (*Satorra & Bentler, 2001*). Furthermore, as recommended by recent literature (*Bonifay et al., 2015*; *Rodriguez, Reise & Haviland, 2016*), we tested whether our bifactor model was essentially unidimensional, calculating in detail: (i) the percentage of uncontaminated correlations (PUCs)—the percentage of correlations that reflect the variance of the general factor (*Bonifay et al., 2015*; *Reise, Bonifay & Haviland, 2013*)—and (ii) the explained common variance (ECV)—the percent of common variance that is attributed to the general factor in the bifactor model (*Reise,*

*Moore & Haviland, 2010*). If a bifactor structure explains the data well, PUCs and ECV provide information about specifying a structural equation model (SEM) measurement model as unidimensional or multidimensional (*Rodriguez, Reise & Haviland, 2016*). The ECV is a measure of the degree of essential unidimensionality (*Sijtsma, 2009*; *Rodriguez, Reise & Haviland, 2016*). It is recommended to take into account the overall data structure and examine PUCs in addition to ECV. *Reise, Bonifay & Haviland (2013)* propose that values of $\omega h$ >.70, ECV >.60, and PUCs >.80 indicate a strong general factor, suggesting a unidimensional model specification should not lead to biased estimates. We also verified the absence of Heywood cases using an analysis of residuals and variances.

**Internal consistency.** As many authors argued, the use of Cronbach's $\alpha$ has several limitations (*Revelle & Zinbarg, 2009*; *Sijtsma, 2009*; *Trizano-Hermosilla & Alvarado, 2016*; *McNeish, 2018*) due to the very restrictive assumptions it is based on, and for this reason we decided to assess internal consistency through the McDonald's omega $\omega$ (*Revelle & Zinbarg, 2009*), considering acceptable values higher than .70. In particular, we computed the hierarchical omega $\omega h$, which is recommended for testing the reliability of the bifactor model and expresses the proportion of variance attributable to a single general factor, treating the variability caused by group factors as measurement error (*Zhang et al., 2024*; *Rodriguez, Reise & Haviland, 2016*).

## RESULTS

**Descriptive statistics.** First, after recording the two reversed-key items, we examined the descriptive statistics to inspect the normality of the item distribution and found that the distribution of several items was problematic (see Table 1). We identified 16 items with a median that corresponded to the minimum value (1). Moreover, we found that seven items showed an absolute skewness index greater than two, and eleven items presented an absolute kurtosis greater than 2. Furthermore, the results of Mardia's test confirmed that the assumption of multivariate normality was not supported (*Mardia, 1970*) (Mardia skewness $= 26497.29$ ($p < .001$), Mardia kurtosis $= 41.24$ ($p < .001$)). Then, we ascertained the presence of redundancy within the subscales. We identified two items with an $r$ >.7, both belonging to the subscale "Persuadability" ("A logical argument can make me change my mind" and "I can be convinced by a good argument").

We noted that two items of the Peer Conformity's subscale showed a weak correlation with the total score (corrected $r < .25$) and that the subscale "Sensation Contagion" was almost entirely composed of items with a weak discriminative power—since they had a median corresponding to the minimum or maximum value—and/or with absolute skewness and/or kurtosis strongly deviated from the norm and/or did not contribute to the Cronbach's alpha. For this reason, we decided to remove this subscale from the successive analyses. We also removed the two items that presented a weak correlation with the total score ("I dress very differently from my friends" and "I don't like most of the movies, my friends, like"). Moreover, we eliminated the two redundant items, the one that slightly seemed more difficult to understand ("A logical argument can make me change my mind"). Finally, we removed three other items that presented excessive skewness and

**Table 1  Descriptive statistics of items of the MISS of Study 1.**

|        | Mean | SD   | Median | Minimum | Maximum | Skewness | Kurtosis |
|--------|------|------|--------|---------|---------|----------|----------|
| MISS1  | 2.14 | 0.91 | 2      | 1       | 5       | 0.72     | 0.05     |
| MISS2  | 2.38 | 0.90 | 2      | 1       | 5       | 0.23     | −0.31    |
| MISS3  | 2.38 | 0.98 | 2      | 1       | 5       | 0.35     | −0.37    |
| MISS4  | 2.22 | 0.98 | 2      | 1       | 5       | 0.75     | 0.21     |
| MISS5  | 1.43 | 0.76 | 1      | 1       | 5       | 2.02     | 4.48     |
| MISS6  | 1.99 | 0.79 | 2      | 1       | 5       | 0.62     | 0.33     |
| MISS7  | 2.32 | 0.94 | 2      | 1       | 5       | 0.38     | −0.21    |
| MISS8  | 1.96 | 0.84 | 2      | 1       | 5       | 0.58     | −0.14    |
| MISS9  | 1.72 | 0.85 | 2      | 1       | 5       | 1.21     | 1.52     |
| MISS10 | 1.39 | 0.66 | 1      | 1       | 4       | 1.76     | 2.81     |
| MISS11 | 1.46 | 0.73 | 1      | 1       | 5       | 1.74     | 3.13     |
| MISS12 | 3.21 | 0.97 | 3      | 1       | 5       | 0.08     | −0.27    |
| MISS13 | 3.19 | 0.94 | 3      | 1       | 5       | 0.04     | −0.08    |
| MISS14 | 2.87 | 0.92 | 3      | 1       | 5       | −0.15    | −0.22    |
| MISS15 | 2.07 | 0.85 | 2      | 1       | 4       | 0.50     | −0.31    |
| MISS16 | 2.85 | 1.02 | 3      | 1       | 5       | 0.09     | −0.36    |
| MISS17 | 1.90 | 0.92 | 2      | 1       | 5       | 1.02     | 0.76     |
| MISS18 | 2.44 | 0.98 | 2      | 1       | 5       | 0.33     | −0.35    |
| MISS19 | 3.01 | 0.95 | 3      | 1       | 5       | −0.04    | −0.31    |
| MISS20 | 2.10 | 0.99 | 2      | 1       | 5       | 0.73     | 0.05     |
| MISS21 | 2.30 | 0.98 | 2      | 1       | 5       | 0.62     | 0.02     |
| MISS22 | 1.76 | 0.84 | 2      | 1       | 5       | 1.22     | 1.63     |
| MISS23 | 2.04 | 0.90 | 2      | 1       | 5       | 0.74     | 0.34     |
| MISS24 | 2.38 | 0.99 | 2      | 1       | 5       | 0.34     | −0.35    |
| MISS25 | 2.12 | 0.84 | 2      | 1       | 4       | 0.38     | −0.44    |
| MISS26 | 1.17 | 0.43 | 1      | 1       | 4       | 2.84     | 9.12     |
| MISS27 | 1.99 | 1.04 | 2      | 1       | 5       | 0.95     | 0.35     |
| MISS28 | 1.47 | 0.72 | 1      | 1       | 5       | 1.81     | 4.05     |
| MISS29 | 1.31 | 0.65 | 1      | 1       | 5       | 2.57     | 7.85     |
| MISS30 | 1.65 | 0.96 | 1      | 1       | 5       | 1.53     | 1.67     |
| MISS31 | 1.30 | 0.63 | 1      | 1       | 5       | 2.34     | 5.90     |
| MISS32 | 1.39 | 0.72 | 1      | 1       | 5       | 2.00     | 3.97     |
| MISS33 | 1.34 | 0.62 | 1      | 1       | 4       | 1.75     | 2.44     |
| MISS34 | 1.99 | 0.96 | 2      | 1       | 5       | 0.87     | 0.42     |
| MISS35 | 1.27 | 0.57 | 1      | 1       | 4       | 2.30     | 5.36     |
| MISS36 | 1.52 | 0.75 | 1      | 1       | 5       | 1.37     | 1.53     |
| MISS37 | 1.23 | 0.55 | 1      | 1       | 5       | 3.04     | 11.79    |
| MISS38 | 2.90 | 1.18 | 3      | 1       | 5       | 0.15     | −0.74    |
| MISS39 | 3.47 | 1.11 | 3      | 1       | 5       | −0.11    | −0.82    |
| MISS40 | 2.66 | 1.17 | 3      | 1       | 5       | 0.24     | −0.82    |
| MISS41 | 2.98 | 1.14 | 3      | 1       | 5       | 0.08     | −0.67    |
| MISS42 | 2.89 | 1.26 | 3      | 1       | 5       | 0.16     | −0.96    |

**Table 1** (*continued*)

|  | Mean | SD | Median | Minimum | Maximum | Skewness | Kurtosis |
|---|---|---|---|---|---|---|---|
| MISS43 | 2.79 | 1.17 | 3 | 1 | 5 | 0.13 | −0.83 |
| MISS44 | 3.04 | 1.17 | 3 | 1 | 5 | −0.01 | −0.76 |
| MISS45 | 2.54 | 1.13 | 3 | 1 | 5 | 0.36 | −0.54 |
| MISS46 | 3.11 | 1.20 | 3 | 1 | 5 | 0.04 | −0.87 |
| MISS47 | 3.24 | 1.24 | 3 | 1 | 5 | −0.12 | −0.92 |
| MISS48 | 3.52 | 1.13 | 4 | 1 | 5 | −0.30 | −0.71 |
| MISS49 | 3.03 | 1.19 | 3 | 1 | 5 | 0.03 | −0.86 |
| MISS50 | 3.65 | 1.14 | 4 | 1 | 5 | −0.41 | −0.69 |
| MISS51 | 2.20 | 0.89 | 2 | 1 | 5 | 0.48 | −0.34 |
| MISS52 | 2.08 | 0.84 | 2 | 1 | 5 | 0.66 | 0.37 |
| MISS53 | 2.42 | 0.85 | 2 | 1 | 5 | 0.12 | −0.33 |
| MISS54 | 3.26 | 1.03 | 3 | 1 | 5 | −0.42 | −0.24 |
| MISS55 | 3.75 | 0.90 | 4 | 1 | 5 | −0.84 | 0.85 |
| MISS56 | 2.21 | 0.94 | 2 | 1 | 5 | 0.40 | −0.53 |
| MISS57 | 1.51 | 0.67 | 1 | 1 | 4 | 1.24 | 1.48 |
| MISS58 | 2.40 | 0.86 | 2 | 1 | 5 | 0.36 | 0.23 |
| MISS59 | 2.15 | 0.95 | 2 | 1 | 5 | 0.76 | 0.32 |
| MISS60 | 1.79 | 0.78 | 2 | 1 | 4 | 0.75 | 0.11 |
| MISS61 | 1.72 | 0.76 | 2 | 1 | 5 | 1.12 | 1.78 |
| MISS62 | 1.88 | 0.89 | 2 | 1 | 5 | 0.72 | −0.13 |
| MISS63 | 1.57 | 0.80 | 1 | 1 | 5 | 1.43 | 1.92 |
| MISS64 | 1.52 | 0.65 | 1 | 1 | 4 | 1.00 | 0.41 |

**Notes.**

SD, standard deviation.

kurtosis and had a median coincident with one of the extremes ("Sometimes I want a product because I like the person endorsing it", "I get my style from certain celebrities" and "I use advertisements as a guide for shopping").

We verified the absence of Heywood cases using the analysis of residuals and variances, and then we examined the modification indices to understand how to improve the model through the removal of other items and/or the addition of covariances between the residuals, where theoretically justifiable. Cross-loadings were admitted in three cases (*e.g.*, we allowed the item "Commercials sometimes make me want products that I did not know I needed" to reflect the "Persuadability" construct, as well as "Consumer"). In the end, we arrived at a forty-six-item reduced version of the scale.

**Model estimation and evaluation.** As reported in Table 2 the values of the fit indices for all the tested models. The one-factor model revealed inadequate, and all the fit indices considered were below the recommended values (CFI = 0.86, TLI = 0.85, SRMR = .13, RMSEA = .14, $\chi 2/df$ = 8.07). The four factors and the higher-order models yielded acceptable fit and did not show appreciable differences, but we decided to concentrate subsequent analysis on the bifactor model since this one was more theoretically motivated and showed better fit (CFI = 0.98, TLI = 0.97, SRMR = .07, RMSEA = .06, $\chi 2/df$ = 2.25). The $\chi 2$ difference test also revealed that the bifactor model had a significantly better fit than the other models (the output of the model comparison is shown in Table 3). Values

**Table 2  Fit indices for CFA on the four models of Study 1.**

|  | Chisq | df | $\chi^2$/df | CFI | TLI | SRMR | RMSEA |
|---|---|---|---|---|---|---|---|
| One factor | 7,979.754 | 989 | 8.07 | 0.86 | 0.85 | 0.13 | 0.14 |
| Four factors | 2,442.825 | 983 | 2.48 | 0.97 | 0.97 | 0.07 | 0.07 |
| Hierarchical | 2,504.348 | 985 | 2.54 | 0.97 | 0.97 | 0.07 | 0.07 |
| Bifactor | 2,121.367 | 943 | 2.25 | 0.98 | 0.97 | 0.07 | 0.06 |

Notes.

Chisq, Chi-squared test; df, degrees of freedom; $\chi^2$/df, ratio of chi-squared to degrees of freedom; CFI, Comparative Fit Index; TLI, Tucker-Lewis Index; RMSEA, Root Mean Square Error of Approximation; SRMR, Standardized Root Mean Squared Residual.

**Table 3  Comparison between models of Study 1.**

|  | df | Chisq | Chisq diff | Df diff | Pr(>Chisq) |
|---|---|---|---|---|---|
| Bifactor | 943 | 2,121.4 |  |  |  |
| Four factors | 983 | 2,442.8 | 321.5 | 40 | <.001 |
| Hierarchical | 985 | 2,504.3 | 61.5 | 2 | <.001 |
| One factor | 989 | 7,979.8 | 5,475.4 | 4 | <.001 |

Notes.

df, degrees of freedom; Chisq, chi-squared test; Chisq diff, chi-squared test difference; Df diff, degrees of freedom difference; Pr(>Chisq), *p*-value.

of PUCs and ECV for the bifactor model were respectively 0.76 and 0.47 and did not reach the suggested cutoffs (PUCs >0.80 and ECV >0.60) to consider the MISS as essentially unidimensional.

**Internal consistency.** The reliability in terms of internal consistency showed a $\omega h$ of .77 for the suggestibility factor. The group (specific) McDonald's Omega of the subscales ranged from .37 to .54 (Consumer Suggestibility = .48; Persuadability = .54; Physiological Reactivity = .43; Peer Conformity = .37).

## Study 2. Cross-validation study
### Participants
A total of 277 participants (196 females, 71%), aged 18 to 73 years (mean age = 30.56 years; SD = 12.58), completed the new version of the scale. Ten participants (2%) received a secondary school education, 144 (52%) a high school diploma or equivalent, 61 (22%) a bachelor's degree, 46 (17%) a master's degree, and 15 (5%) a doctorate. The sample consisted of 139 students (50%), 68 employees (25%), 39 professional workers (14%), 23 unemployed (8%), and eight workers (3%). All subjects, informed about the purpose of the study, gave their written consent. They were recruited from universities and communities through local advertisements.

### Procedure
Participants received an online version of the scales prepared through *Google Forms*. After providing informed consent, we reminded the participants that they were allowed to quit the study at any point. In addition to personal data collection, we asked participants to complete the following scales: the MISS, the Ten Item Personality Inventory (TIPI; *Chiorri et al., 2015*), the HEXACO-60 (*Ashton & Lee, 2009*), and the Self-Monitoring Scale

(SMS; *Snyder, 1974*). The TIPI and the HEXACO-60 focused on personality traits, while the SMS explored self-monitoring ability. These measures were used to test divergent validity, investigating whether the MISS is measuring a different construct. Participants were required to answer all the questions. While the order of the scales was kept constant, the order of the items within each test was randomized across participants.

## Measures

### The ten item personality inventory (*Chiorri et al., 2015*)

The ten item personality inventory (TIPI) is a measure of the Five Factor Model that can be administered in a few minutes and consists of ten items, two for each dimension (Extraversion, Agreeableness, Conscientiousness, Neuroticism, and Openness to Experience). Items used a 7-point Likert scale ranging from 1 (*strongly disagree*) to 7 (*strongly agree*), and there is a reverse item for each dimension. Examples of items are "extraverted, enthusiastic" and "reserved, quiet" for Extraversion; "dependable, self-disciplined" and "disorganized, careless" for Conscientiousness.

### The HEXACO-60 (*Ashton & Lee, 2009*)

The HEXACO-60 is a short personality inventory of the six major dimensions of personality: Honesty-Humility (*e.g.*, "I wouldn't pretend to like someone just to get that person to do favors for me"), Emotionality, Extraversion, Agreeableness, Conscientiousness (*e.g.*, "I plan ahead and organize things, to avoid scrambling at the last minute"), and Openness to Experience. This scale consists of 60 items on a 5-point Likert scale ranging from 1 (*strongly disagree*) to 5 (*strongly agree*). There are also some reverse items.

### The self-monitoring scale (*Snyder, 1974*)

The self-monitoring scale (SMS) was developed by *Snyder (1974)* and measures individual differences in self-monitoring, that is the extent to which individuals monitor, observe, and control their self-presentation and expressive behavior in accordance with social cues that indicate socially approved behaviors. This scale consists of 25 items and the responses are summed to form an overall score of self-monitoring. Examples of items are "I find it hard to imitate the behavior of other people", and "I'm not always the person I appear to be".

## Data analysis

**Descriptive statistics.** Statistical analyses were performed with R (*R Core Team, 2022*). We followed the same analytical approach as in the first study. The assumption of multivariate normality was tested through the test of Mardia, where *p*-values greater than .05 indicate the assumption is met.

**Factorial structure.** We assessed the dimensionality of the MISS through confirmatory factor analysis (CFA), using the R package *lavaan* (*Rosseel, 2012*), and testing the same models of the first study. As in the first study, we tested four measurement models: (1) one unidimensional (for comparison only), (2) another with four factors (the four subscales of the MISS), (3) a hierarchical second-order latent variable model (SOLV) with the same four factors as first-order factors, and a second-order "Suggestibility" factor, and (4) a bifactor model.

**Model estimation and evaluation.** The goodness of fit of the models was evaluated using the following indices: ratio of chi-squared to degrees of freedom ($\chi2/df$); RMSEA, CFI, TLI, and SRMR. Values below 0.10 indicated a good fit for RMSEA and SRMR, while for CFI and TLI values above 0.90 were considered acceptable (*Hu & Bentler, 1999*). Values of $\chi2/df$ <3 indicated acceptable fit. The best model was selected through (i) an evaluation of the fit indices; and (ii) the $\chi2$ difference test between models, as in the first study. Moreover, we computed the PUCs and ECV indices for the bifactor model, with the aim of investigating whether the model was essentially unidimensional. Values of PUCs >.80 and ECV >.60 suggest the presence of unidimensionality.

**Internal consistency.** We used McDonald's hierarchical Omega (*McDonald, 1999*) as an internal reliability index and considered acceptable values such as 0.70 or higher.

**Divergent validity.** To assess divergent validity we used the Self-Monitoring Scale (*Snyder, 1974*). We have modeled, using an SEM framework, the relationship between the two latent variables including the respective measurement models. The MISS measurement model was a bifactor model where all items loaded on the general factor suggestibility, and specific items loaded also only on another specific factor. For the Self-Monitoring Scale, we used an unidimensional model where all the items reflected the general construct. We fixed the variance of the latent variables at 1 and verified that the upper limit of the 95% confidence intervals of the correlation between the latent variables did not exceed the values defined as problematic (.8) by the *Rönkkö & Cho (2022)* simulation study. Furthermore, we calculated the correlations between the MISS total score, its subscales, and two self-report measures of personality, the TIPI and the HEXACO-60, to ascertain whether the MISS measures a different construct.

## RESULTS

**Descriptive statistics.** Descriptive statistics of the items are reported in Table 4.

First, we assessed multivariate normality using the Mardia's test, but results showed that the assumption was not supported (Mardia skewness = 23714.10 ($p < .001$), Mardia kurtosis =28.98 ($p < .001$)). Then, we inspected the means and medians of items and found that no one was falling into an extreme value (1 or 5). We also checked items skewness and kurtosis and there were no items with values greater than 2. Moreover, we checked item correlations within the subscales and found there were no correlations higher than .7, confirming the absence of redundancy within the subscales.

**Model estimation and evaluation.** We performed a CFA through the *lavaan* library (*Rosseel, 2012*) to verify the structure validity of the scale. Since the assumption of multivariate normality was not met, we used a diagonally weighted least square estimator (DWLS).

The results of fit indices for all models are reported in Table 5. The overall fit statistics suggest a poor fit for the unidimensional solution (CFI = 0.87, TLI = 0.86, SRMR = .11, RMSEA = .12, $\chi2/df$ = 4.96). The four factors and the four factors hierarchical models showed similar results and acceptable fit. The bifactor model performed better than the other models and the overall fit statistics suggested a good fit (CFI = 0.97, TLI = 0.97,

**Table 4 Descriptive statistics of items of the MISS of Study 2.**

|         | Mean | SD   | Median | Minimum | Maximum | Skewness | Kurtosis |
|---------|------|------|--------|---------|---------|----------|----------|
| MISS 1  | 2.30 | 1.02 | 2      | 1       | 5       | 0.52     | −0.12    |
| MISS 2  | 2.37 | 0.94 | 2      | 1       | 5       | 0.16     | −0.55    |
| MISS 3  | 2.39 | 0.89 | 2      | 1       | 5       | 0.24     | −0.25    |
| MISS 4  | 2.43 | 0.92 | 2      | 1       | 5       | 0.47     | 0.01     |
| MISS 5  | 2.35 | 0.85 | 2      | 1       | 5       | 0.42     | 0.23     |
| MISS 6  | 2.70 | 1.01 | 3      | 1       | 5       | 0.29     | −0.25    |
| MISS 7  | 2.02 | 0.89 | 2      | 1       | 5       | 0.66     | 0.16     |
| MISS 8  | 2.08 | 0.84 | 2      | 1       | 5       | 0.67     | 0.61     |
| MISS 9  | 3.51 | 0.87 | 3      | 1       | 5       | −0.18    | −0.06    |
| MISS 10 | 3.22 | 0.84 | 3      | 1       | 5       | 0.05     | 0.07     |
| MISS 11 | 2.38 | 0.88 | 2      | 1       | 5       | 0.31     | −0.19    |
| MISS 12 | 3.22 | 1.03 | 3      | 1       | 5       | −0.14    | −0.51    |
| MISS 13 | 2.23 | 0.91 | 2      | 1       | 5       | 0.43     | −0.05    |
| MISS 14 | 2.78 | 0.87 | 3      | 1       | 5       | 0.13     | 0.05     |
| MISS 15 | 3.75 | 0.83 | 4      | 1       | 5       | −0.52    | 0.38     |
| MISS 16 | 2.58 | 1.13 | 2      | 1       | 5       | 0.47     | −0.42    |
| MISS 17 | 2.78 | 0.95 | 3      | 1       | 5       | 0.22     | −0.14    |
| MISS 18 | 2.12 | 0.90 | 2      | 1       | 5       | 0.60     | −0.19    |
| MISS 19 | 2.24 | 0.91 | 2      | 1       | 5       | 0.29     | −0.46    |
| MISS 20 | 2.65 | 1.08 | 3      | 1       | 5       | 0.02     | −0.70    |
| MISS 21 | 2.47 | 0.89 | 2      | 1       | 5       | 0.30     | −0.01    |
| MISS 22 | 3.01 | 1.29 | 3      | 1       | 5       | 0.02     | −1.06    |
| MISS 23 | 4.02 | 1.04 | 4      | 1       | 5       | −0.81    | −0.15    |
| MISS 24 | 2.72 | 1.19 | 3      | 1       | 5       | 0.17     | −0.97    |
| MISS 25 | 3.30 | 1.00 | 3      | 1       | 5       | −0.16    | −0.48    |
| MISS 26 | 3.58 | 1.20 | 4      | 1       | 5       | −0.47    | −0.74    |
| MISS 27 | 3.34 | 1.05 | 3      | 1       | 5       | −0.19    | −0.62    |
| MISS 28 | 3.45 | 1.09 | 3      | 1       | 5       | −0.38    | −0.40    |
| MISS 29 | 2.96 | 1.13 | 3      | 1       | 5       | −0.03    | −0.73    |
| MISS 30 | 3.60 | 1.02 | 4      | 1       | 5       | −0.37    | −0.27    |
| MISS 31 | 3.70 | 1.01 | 4      | 1       | 5       | −0.52    | −0.09    |
| MISS 32 | 3.83 | 0.95 | 4      | 1       | 5       | −0.56    | −0.03    |
| MISS 33 | 3.57 | 1.03 | 4      | 1       | 5       | −0.37    | −0.44    |
| MISS 34 | 4.15 | 0.94 | 4      | 1       | 5       | −1.03    | 0.62     |
| MISS 35 | 2.46 | 0.92 | 2      | 1       | 5       | 0.12     | −0.46    |
| MISS 36 | 2.57 | 0.93 | 3      | 1       | 5       | 0.15     | −0.25    |
| MISS 37 | 3.10 | 0.89 | 3      | 1       | 5       | −0.27    | 0.22     |
| MISS 38 | 2.49 | 0.92 | 2      | 1       | 5       | 0.16     | −0.49    |
| MISS 39 | 1.72 | 0.74 | 2      | 1       | 5       | 1.08     | 1.98     |
| MISS 40 | 2.96 | 0.82 | 3      | 1       | 5       | 0.04     | 0.68     |
| MISS 41 | 2.48 | 0.99 | 2      | 1       | 5       | 0.15     | −0.66    |
| MISS 42 | 2.12 | 0.94 | 2      | 1       | 5       | 0.64     | 0.05     |

**Table 4** (*continued*)

|  | Mean | SD | Median | Minimum | Maximum | Skewness | Kurtosis |
|---|---|---|---|---|---|---|---|
| MISS 43 | 2.18 | 0.83 | 2 | 1 | 5 | 0.31 | −0.09 |
| MISS 44 | 2.29 | 1.04 | 2 | 1 | 5 | 0.41 | −0.57 |
| MISS 45 | 1.92 | 0.92 | 2 | 1 | 5 | 0.79 | 0.16 |
| MISS 46 | 1.94 | 0.85 | 2 | 1 | 5 | 0.84 | 0.61 |

**Notes.**

SD, standard deviation.

**Table 5 Fit indices for CFA on the four models of Study 2.**

|  | Chisq | df | $\chi^2/df$ | CFI | TLI | SRMR | RMSEA |
|---|---|---|---|---|---|---|---|
| One factor | 4,911.104 | 989 | 4.96 | 0.87 | 0.86 | 0.11 | 0.12 |
| Four factors | 2,575.231 | 983 | 2.62 | 0.95 | 0.94 | 0.08 | 0.08 |
| Hierarchical | 2,649.092 | 985 | 2.69 | 0.94 | 0.94 | 0.08 | 0.08 |
| Bifactor | 1,878.551 | 943 | 1.99 | 0.97 | 0.97 | 0.07 | 0.06 |

**Notes.**

Chisq, Chi-squared test; df, degrees of freedom; $\chi^2/df$, ratio of chi-squared to degrees of freedom; CFI, Comparative Fit Index; TLI, Tucker-Lewis Index; RMSEA, Root Mean Square Error of Approximation; SRMR, Standardized Root Mean Squared Residual.

**Table 6 Comparison between models of Study 2.**

|  | df | Chisq | Chisq diff | Df diff | Pr(>Chisq) |
|---|---|---|---|---|---|
| Bifactor | 943 | 1,878.6 |  |  |  |
| Four factors | 983 | 2,575.2 | 696.68 | 40 | <.001 |
| Hierarchical | 985 | 2,649.1 | 73.86 | 2 | <.001 |
| One factor | 989 | 4,911.1 | 2,262.01 | 4 | <.001 |

**Notes.**

df, degrees of freedom; Chisq, chi-squared test; Chisq diff, chi-squared test difference; Df diff, degrees of freedom difference; Pr(>Chisq), *p*-value.

SRMR = .07, RMSEA = .06, $\chi^2/df$ = 1.99). Then, we compared the four models, and from the results of the $\chi^2$ difference test (reported in Table 6), the bifactor model outperformed the other three models. Then, we computed PUCs and ECV for the bifactor model and their values (respectively 0.76 and 0.54) were below the indicated cutoffs, suggesting a multidimensional structure of the MISS.

To better understand how variance is partitioned, we explored the loadings of the bifactor solution for the general factor suggestibility and the four subscales (results are reported in Table 7). The average loading on the general factor is .46, while those of the subscales are .46 for consumer .22 for persuadability, .49 for physical reactivity, and .40 for conformity. These data show that the general factor explains about 20% of the variance in the items. The subscales consumer, physical reactivity, and conformity explain, over and above the general factor, on average a similar part of the variance, while the subscale persuadability is only 5%. In Table 7 we put in bold the items loading on the general factor greater than .50. These items provide the best discrimination on the suggestibility factor. On the other hand, the grouping factor items loading greater than the general factor loadings (also in bold) could be considered relatively better measures of the specific subscale construct than of

suggestibility. Among the subscales, the physical reactivity one is the only one that appears a bit problematic, since nine of the 13 items have greater loadings on the specific grouping factor than on the general one. This could derive from a measurement problem or from the fact that it may measure something separate from the suggestibility construct.

**Internal consistency.** The scale demonstrated good internal consistency, with a McDonald's hierarchical Omega of .81 for the general factor suggestibility and values ranging from .28 to .72 for the subscales (Consumer Suggestibility = .50 ; Persuadability = .72; Physiological Reactivity = .28; Peer Conformity = .57).

**Divergent validity.** A SEM was used to investigate the relationship between the MISS and the SMS. Results highlighted that the two measures, once accounted for the subscales contribution, are strongly correlated (Estimate = 0.72, se = 0.04, $z$ = 17.60, $p < .001$). The upper limit of the confidence interval (95% CI [0.641–0.801]) barely exceeded the threshold (.8) suggested by *Rönkkö & Cho (2022)*. On the other hand, the MISS was not significantly correlated with any dimension of the HEXACO-60, nor the TIPI (results are reported in Table 8).

# DISCUSSION

The Multidimensional Iowa Suggestibility Scale (*Kotov, Bellman & Watson, 2004*) is a self-report measure of individual differences in suggestibility, which covers different domains, such as conformity, persuadability, physical reactivity, and consumer. Our study aimed to adapt the scale in a large sample of Italian healthy individuals, investigating its factor structure and psychometric properties.

In the first study, we examined item characteristics of the original five subscales and dropped all those that resulted problematic, redundant, or not discriminative. We also removed the subscale "Sensation Contagion" because it had a weak discriminative power. Reliability analyses highlighted good internal consistency with McDonald's hierarchical omega, and this result is consistent with that reported in the original validation of the scale by *Kotov, Bellman & Watson (2004)*. The confirmatory factor analysis showed good fit indices for the bifactor model. We also compared the bifactor model with the other three models, and the bifactor model outperformed the others. This is due to the fact that it allows us to better explain how variance is partitioned between the general factor suggestibility and the subscales (*Reise, Moore & Haviland, 2010*).

We conducted a second study to cross-validate the new version of the MISS. We administered the scale to a new independent sample to investigate whether the factor structure and reliability were confirmed. This would strengthen and validate the results we found in the first study, also increasing generalizability. As shown in the results, we found that the MISS demonstrated good psychometric properties. Reliability in terms of internal consistency was good. With regard to factorial structure, we tested four measurement models, as in the first study. The first was unidimensional, and suggestibility represented the general factor. We tested this model for comparison only since we already knew from the original validation study that there was evidence of multidimensionality. The second model was a four-factor model, where each subscale represented a factor. The third model was a

**Table 7  Factor loadings on general and specific grouping factors of bifactor solution of Study 2.**

|         | Sugg | Cons | Persu | PR  | Conf |
|---------|------|------|-------|-----|------|
| MISS1   | **.56** | .56  |       |     |      |
| MISS2   | .44  | **.56** |       |     |      |
| MISS3   | **.54** | .56  |       |     |      |
| MISS4   | **.63** | .17  |       |     |      |
| MISS5   | **.51** | .46  |       |     |      |
| MISS6   | .48  | **.49** |       |     |      |
| MISS7   | **.52** | .45  |       |     |      |
| MISS8   | **.52** | .47  |       |     |      |
| MISS9   | .43  |      | .28   |     |      |
| MISS10  | **.51** |      | .32   |     |      |
| MISS11  | **.59** |      | .39   |     |      |
| MISS12  | .30  |      | .24   |     |      |
| MISS13  | **.55** |      | .36   |     |      |
| MISS14  | .45  |      | .44   |     |      |
| MISS15  | .39  |      | .07   |     |      |
| MISS16  | .32  |      | .25   |     |      |
| MISS17  | **.60** |      | −.14  |     |      |
| MISS18  | **.65** |      | .23   |     |      |
| MISS19  | **.61** |      | .28   |     |      |
| MISS20  | **.53** |      | −.11  |     |      |
| MISS21  | **.63** |      | .36   |     |      |
| MISS22  | .35  |      |       | .34 |      |
| MISS23  | .21  |      |       | **.72** |      |
| MISS24  | .39  |      |       | **.47** |      |
| MISS25  | .42  |      |       | **.48** |      |
| MISS26  | .26  |      |       | **.40** |      |
| MISS27  | .46  |      |       | .20 |      |
| MISS28  | .27  |      |       | **.55** |      |
| MISS29  | .45  |      |       | .44 |      |
| MISS30  | .30  |      |       | **.50** |      |
| MISS31  | .43  |      |       | .42 |      |
| MISS32  | .25  |      |       | **.50** |      |
| MISS33  | .19  |      |       | **.59** |      |
| MISS34  | .22  |      |       | **.79** |      |
| MISS35  | .44  |      |       |     | **.68** |
| MISS36  | .46  |      |       |     | **.63** |
| MISS37  | .42  |      |       |     | .31  |
| MISS38  | .39  |      |       |     | **.46** |
| MISS39  | **.70** |      |       |     | .32  |
| MISS40  | .45  |      |       |     | .35  |
| MISS41  | .41  |      |       |     | **.62** |
| MISS42  | **.52** |      |       |     | .55  |

**Table 7** (*continued*)

|  | Sugg | Cons | Persu | PR | Conf |
|---|---|---|---|---|---|
| MISS43 | **.55** |  |  |  | .52 |
| MISS44 | .48 |  |  |  | .14 |
| MISS45 | **.61** |  |  |  | −.03 |
| MISS46 | **.66** |  |  |  | .22 |

**Notes.**

Boldface shows a general factor loading greater than .50, and group factor loadings greater than general factor loadings.
Sugg, suggestibility total score; Conf, MISS conformity subscale; Cons, MISS consumer subscale; PR, MISS physical reactivity subscale; Pers, MISS persuadability subscale.

**Table 8  Correlation matrix between MISS, its subscales, TIPI, Hexaco-60 and Self-Monitoring Scale of Study 2.**

|  | Conf | Cons | PR | Pers | MISS | SELF | HEXACO | TIPI |
|---|---|---|---|---|---|---|---|---|
| Conf | – |  |  |  |  |  |  |  |
| Cons | 0.44*** | – |  |  |  |  |  |  |
| PR | 0.25*** | 0.35*** | – |  |  |  |  |  |
| Pers | 0.54*** | 0.48*** | 0.42*** | – |  |  |  |  |
| MISS | 0.71*** | 0.71*** | 0.71*** | 0.81*** | – |  |  |  |
| SELF | 0.33*** | 0.30*** | 0.09 | 0.34*** | 0.35*** | – |  |  |
| HEXACO-60 | −0.15 | 0.01 | 0.21*** | −0.10 | 0.00 | −0.33*** | – |  |
| TIPI | 0.02 | 0.11 | 0.17 | −0.10 | 0.05 | 0.05 | NA | – |

**Notes.**

Conf, MISS conformity subscale; Cons, MISS consumer subscale; PR, MISS physical reactivity subscale; Pers, MISS persuadability subscale; MISS, MISS total score; SELF, Self-Monitoring Scale; TIPI, Ten Item Personality Inventory Scale; Hexaco-60, Hexaco-60 personality Scale.

*** $p < .001$

hierarchical second-order latent variable model (SOLV), with the subscales as first-order factors, and suggestibility as second-order factor. The fourth was a bifactor model, in which all items loaded on the general factor suggestibility and also on a specific grouping factor. CFAs revealed that the bifactor structure was confirmed, and this is consistent with the results found in our first study. This is an important point to discuss since the use of the bifactor model allowed us to investigate and understand the contribution of general and specific factors more deeply than could be done with other models, and for this reason, it should not be considered only as an alternative model to test against (*Dunn & McCray, 2020*). A deeper analysis of $\omega h$, PUCs, and ECV permitted us to further explore the dimensionality of the measurement model of the MISS. However, results showed mixed evidence since, on the one hand, the high $\omega h$ suggests the presence of a relatively strong general factor, but on the other hand PUCs and ECV values did not reach the cut-offs recommended for essential unidimensionality, thus indicating a multidimensional structure of the scale. At this point, we decided to explore in detail the factor loadings on the general and the specific grouping factors to further understand the contribution of each and, in particular, the adequacy of reporting a total or composite score. We observed that on average the general factor suggestibility explained about the 20% of shared variance. The subscales accounted for a similar or lower level of explained variance, apart from the

contribution of suggestibility. The loadings of the subscales were tendentially lower than those of the general factor, except for the physical reactivity subscale.

Given these results, and analyzed the items' content, we concluded that there was no justification to report only the separate scores for the subscales. Moreover, as argued by *Sinharay, Haberman & Puhan (2007)*, we are convinced that the total score is more reliable compared to the composite score and constitutes a better predictor of the true individual's score on suggestibility traits.

The Italian adaptation of the MISS demonstrated acceptable construct validity. We tested divergent validity and found that there was a lack of correlation between the MISS score and the TIPI and HEXACO-60 scores and a moderate correlation between the MISS score and the Self-Monitoring scale. However, when we modeled with SEM the relationship between the general factor of the MISS and the factor of the SMS, once accounted for the specific factors' contribution, the correlation became stronger and the upper limit of the confidence interval fell just above the threshold considered marginally problematic by *Rönkkö & Cho (2022)*. So, it remains unclear whether the MISS is measuring a different construct from self-monitoring.

In the end, the MISS revealed good psychometric properties and can represent a precious tool for a wide evaluation of individual differences in suggestibility in many domains. Future studies are needed to confirm all these findings and maybe test the scale, or a revised version, in clinical samples.

### Limitations

This study also has some important limitations. First, we did not test convergent validity. Future studies could fill this gap, using the Harvard Group Scale of Hypnotic Susceptibility, Form A. Second, the sample is not strongly representative of the entire Italian population, and there is a strong prevalence of women and students.

## CONCLUSION

To date, in Italy, the most used scale to measure suggestibility is the Italian version of the Harvard Group Scale of Hypnotic Susceptibility, Form A (*De Pascalis, Russo & Marucci, 2000*). More instruments are needed. To fill this gap, we adapted the Multidimensional Iowa Suggestibility Scale (MISS) in a large sample of Italian individuals and the scale showed good psychometric properties. Measuring individual differences in suggestibility is crucial, in many different domains, such as clinical psychology, neuromarketing, and neuropharmacology. In particular, the objective assessment of suggestibility could become of interest to pharmacological studies assessing the placebo effect. When an inert substance is delivered as a standard treatment and the clinical outcome is improved, this phenomenon is known as the placebo effect. One of the main clinical conditions for which the rate of placebo response is high is Parkinson's disease (*Quattrone et al., 2018*). This phenomenon is drastically driven by verbal suggestions that induce an indirect overactivation of the reward network compensating the dopaminergic deficit in patients with Parkinson's disease. We believe that the adaptation of this scale could be of interest for clinical practice to assess the probability of neurological patients being prone to placebo effects.

## ACKNOWLEDGEMENTS

We thank Maria Gioia Chiatante, Silvia Galati, Caterina Scavo and Francesco Scarfone for help with data collection.

### Funding

The current research was funded by MUR-PRIN (Ministry of University and Research—Projects of National Relevant Interest) granted to Marco Tullio Liuzza (grant number: 2022TN4ETY). The funders had no role in study design, data collection and analysis, decision to publish, or preparation of the manuscript.

### Grant Disclosures

The following grant information was disclosed by the authors:
MUR-PRIN: 2022TN4ETY.

### Competing Interests

Marco Tullio Liuzza is an Academic Editor for PeerJ. Antonio Cerasa is a scientific consultant for the robotic neurorehabilitation lab of S. Anna Institute in Crotone (Italy), which is a non-academic institution. All the other authors declare that they have no competing interests.

### Author Contributions

- Marco Tullio Liuzza conceived and designed the experiments, analyzed the data, authored or reviewed drafts of the article, funding, and approved the final draft.
- Eva Tolomeo performed the experiments, analyzed the data, prepared figures and/or tables, authored or reviewed drafts of the article, and approved the final draft.
- Giuseppe Occhiuto performed the experiments, analyzed the data, prepared figures and/or tables, authored or reviewed drafts of the article, and approved the final draft.
- Martina Cilurzo performed the experiments, authored or reviewed drafts of the article, and approved the final draft.
- Iolanda Martino conceived and designed the experiments, authored or reviewed drafts of the article, and approved the final draft.
- Antonio Cerasa conceived and designed the experiments, authored or reviewed drafts of the article, and approved the final draft.

### Human Ethics

The following information was supplied relating to ethical approvals (*i.e.*, approving body and any reference numbers):

Ethics Committee of the Calabria Region (Registration protocol n. 285 del 17 Ottobre 2019).

## Data Availability

The datasets, codebooks and R scripts for reproducing the analyses are available at the Open Science Framework (OSF): Tolomeo, Eva, Marco Tullio Liuzza, and Giuseppe Occhiuto. 2023. ''Italian Adaptation of the Multidimensional Iowa Suggestibility Scale (MISS).'' OSF. September 12. doi: 10.17605/OSF.IO/M4P7E.

## Supplemental Information

Supplemental information for this article can be found online at http://dx.doi.org/10.7717/peerj.17145#supplemental-information.

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
