# Peer review of "Italian adaptation of the Multidimensional Iowa Suggestibility Scale (MISS)"

_PeerJ, doi:10.7717/peerj.17145_

## Round 0.1 · original submission · Major Revisions

Please, include any issues that the reviewers have highlighted, or may have missed.

·

Basic reporting

The article is very clearly written and in good English. However, the reference Kotov, Bellman & Watson (2004) is missing from the reference list. It should be added.

The way the MS talks about suggestibility needs some more context. Right now, the MS mainly mentions Barber's scale, which isn't used much today. A better scale to mention would be the Harvard Group Scale of Hypnotic Suggestibility (Shor & Orne, 1962). This is by far the most popular scale currently, and Oakley et al. (2021) have also proposed using it in non hypnotic context. However, some recent studies have criticized it for many reasons, like those by Acunzo & Terhune (2021) and Kallio (2021). Kallio also discusses the history of measuring suggestibility that maight be of help.

The MS needs be clear that MISS scale is not a test of suggestibility. Instead, it's a questionnaire about individuals' perceptions of their own suggestibility based on various statements. While many scales assess suggestibility, often in a hypnotic context, the MISS scale doesn't. This distinction is crucial since the concept of suggestibility was "borne" in association with hypnosis, as noted by Hilgard (1991).

References
Acunzo, D. J., & Terhune, D. B. (2021). A Critical Review of Standardized Measures of Hypnotic Suggestibility. The International journal of clinical and experimental hypnosis, 69(1), 50–71.
Hilgard, E. R. (1991). Suggestibility and suggestions as related to hypnosis. In J. F. Schumaker (Ed.), Human suggestibility: Advances in theory, Research and application
(pp. 37–58). London: Routledge.
Kallio S. (2021). Time to update our suggestibility scales. Consciousness and cognition, 90, 103103.
Shor, R. E., & Orne, E. C. (1962). Harvard group scale of hypnotic susceptibility: Form A. Palo Alto, CA, USA: Consulting Psychologists Press.

Experimental design

The translation of the scale was a good example of how translators should be done - nice to see!

Validity of the findings

Statistics was well done and conclusions well stated.

Additional comments

No comments

·

Basic reporting

It’s great to see translations of the MISS coming out, and I’ve enjoyed reviewing your paper.

However, there are some notable gaps and inaccuracies in your introduction and discussion sections that are important to integrate into your conceptualisation/revisions. I have made some suggestions of sources to review and incorporate but please note this not a comprehensive list of potential resources.

I would encourage the authors to adopt a more up to date definition of suggestions, consider reviewing the following:

Kirsch, I., & Braffman, W. (1999, December). Correlates of hypnotizability: The first empirical study. In Contemporary Hypnosis (Vol. 16, No. 4, pp. 224-230). Chichester, UK: John Wiley & Sons, Ltd..

To address some gaps in the your introduction to the factor structure of suggestibility, please consider reviewing the following:

Polczyk R, Pasek T. Types of suggestibility: Relationships Among Compliance, Indirect, and Direct Suggestibility. International Journal of Clinical and Experimental Hypnosis. 2006;54(4):392-415.

Polczyk, R. (2016). Factor structure of suggestibility revisited: new evidence for direct and indirect suggestibility. Current Issues in Personality Psychology, 4(2), 87-96.

Oakley DA, Walsh E, Mehta MA, Halligan PW, Deeley Q. Direct verbal suggestibility: Measurement and significance. Conscious Cogn. 2021;89:103036. doi:10.1016/j.concog.2020.103036

Barnier AJ, Terhune DB, Polito V, Woody EZ. A componential approach to individual differences in hypnotizability. Psychology of Consciousness: Theory, Research, and Practice. 2022;9(2):130.

In your brief introduction to suggestibility + placebo, it may be helpful to review this study and it’s sources:

Parsons RD, Bergmann S, Wiech K, Terhune DB. Direct Verbal Suggestibility as a Predictor of Placebo Hypoalgesia Responsiveness. Psychosom Med. 2021;83(9):1041-9.

Conclusion:

In your conclusion you state that there is no valid Italian suggestibility test, this is not accurate. Please see the below sources:

De Pascalis, V., Russo, P., & Marucci, F. S. (2000). Italian norms for the Harvard Group Scale of Hypnotic Susceptibility, Form A. The International journal of clinical and experimental hypnosis, 48(1), 44–55. https://doi.org/10.1080/00207140008410360

De Pascalis, V., Bellusci, A., & Russo, P. M. (2000). Italian norms for the Stanford Hypnotic Susceptibility Scale, Form C. The International journal of clinical and experimental hypnosis, 48(3), 315–323. https://doi.org/10.1080/00207140008415249

I would also suggest double checking the reference list and citation formatting. Kotov 2004 is missing from the reference list, and the Sinharay and Puhn citation is incorrect (those are just two examples, I did not thoroughly review for these sorts of errors).

Experimental design

Your experimental design reporting was clear and followed, to my knowledge, standard practices.

Validity of the findings

In your limitations, you highlight that convergent validity can’t be tested due to the lack of a comparable measure. The MISS is reliably associated with a measure of direct verbal suggestibility (the Harvard Group scale); there is an Italian translation of the Harvard:

De Pascalis, V., Russo, P., & Marucci, F. S. (2000). Italian norms for the Harvard Group Scale of Hypnotic Susceptibility, Form A. The International journal of clinical and experimental hypnosis, 48(1), 44–55. https://doi.org/10.1080/00207140008410360

Reviewer 3 ·

Basic reporting

The Authors present a psychometric investigation of the Italian adaptation of the Multidimensional Iowa Suggestibility Scale (MISS) in an Italian sample, testing its factorial structure, divergent validity and internal consistency.

I think the paper is likely to be of interest to all those who use or consider assessing suggestibility in different contexts (e.g. clinical and research).

The manuscript is well written. The writing is clear and comprehensive and the overall study design with the cross-validation component across two studies is good. Overall, the Authors use an appropriate analytical approach in the first study (although - in my opinion - further analyses are needed). In the second study, however, the manuscript suffers from some shortcomings that prevent a proper evaluation of the overall study results (also in the light of the cross-validation approach).

I listed the main critical points that Authors may consider when revising the manuscript. I hope these suggestions will help the Authors to revise the manuscript. For the sake of simplicity, I will report all my comments in this section.

a) Study 1 and Study 2
According to the Authors, in both Study 1 and Study 2, the chi-squared difference test for their four tested models, the bi-factor model is superior to its alternatives; however, a conclusion based on fit indices is not recommended (currently even the g-factor in psychopathology is being questioned).
Following the existing recommendations (i.e., Reise et al., 2013; Bonifay et al., 2015; Bonifay et al., 2017; Rodriguez et al., 2016), I suggest to perform an in-depth diagnosis of the bifactor model by examining - in addition to the hierarchical omega (ωH) - some ancillary indices, including the percentage of uncontaminated correlations (PUCs) and the explained common variance (ECV) (Bonifay et al., 2015; Reise et al., 2013; Reise et al., 2010). Following the suggestions of Reise et al. (2013), ωH values greater than 0.70, PUC values greater than 0.80, and ECV values greater than 0.60 suggest the presence of a general factor (Reise et al., 2013).

I suggest looking at:
Bonifay, W. E., Reise, S. P., Scheines, R., & Meijer, R. R. (2015). When are multidimensional data unidimensional enough for structural equation modeling? An evaluation of the DETECT multidimensionality index. Structural Equation Modeling: A Multidisciplinary Journal, 22(4), 504-516. https://doi.org/10.1080/10705511.2014.938596
Reise, S. P., Moore, T. M., & Haviland, M. G. (2010). Bifactor models and rotations: Exploring the extent to which multidimensional data yield univocal scale scores. Journal of Personality Assessment, 92(6), 544-559. https://doi.org/10.1080/00223891.2010.496477
Reise, S. P., Scheines, R., Widaman, K. F., & Haviland, M. G. (2013). Multidimensionality and structural coefficient bias in structural equation modeling: A bifactor perspective. Educational and Psychological Measurement, 73(1), 5-26. https://doi.org/10.1177/0013164412449831
Olatunji, B. O., Ebesutani, C., & Reise, S. P. (2015). A bifactor model of disgust proneness: Examination of the Disgust Emotion Scale. Assessment, 22(2), 248-262. https://doi.org/10.1177/1073191114541673
Zhang J, Marci T, Marino C, Canale N, Vieno A, Wang J, Chen X. Factorial validity of the problematic social media use scale among Chinese adults. Addict Behav. 2024 Jan;148:107855. doi: 10.1016/j.addbeh.2023.107855.

b) Study 2
- p.16 – “We performed a CFA through the lavaan library (Rosseel, 2012) to verify the structure validity of the scale. Since the assumption of multivariate normality was not met, we used a Maximum Likelihood Robust (MLR) estimator (i.e., estimation with robust standard errors and SatorrañBentler scaled test statistic, see Beaujean 2014)”. Why did the Authors not use the appropriate estimator for non-normal and/or ordinal data (i.e. DWLS) again in Study 2, as in Study 1?

- p.16 (line 317) – "Results are reported in Table 6. The overall fit statistics suggest a poor fit for the unidimensional solution. The four factors and the four factors hierarchical models showed similar results, but still poor fit. On the other hand, the bifactor model performed better and the overall fit statistics suggested a good or acceptable fit.” Looking at Table 6, even the bifactor model has values well below those recommended and, in any case, considered by the Authors for the interpretation of the models. Could this result be due to the use of the MLR to estimate the models? If this is the case, I recommend that the analysis be repeated using the DWLS (in line with the first study).

- p. 17 – “The scale demonstrated good internal consistency, with a McDonald Omega of .79 for the general factor suggestibility and values ranging from .14 to .61 for the subscales”. Are the Authors talking about the hierarchical Omega? If so, it should be specified.

- Relation to external measures – The Authors should be clearer in reporting and explaining results and/or refer to the relevant table where appropriate.

c) Other minor points:
- I suggest using the same terms to refer to the bifactor model (for example, the Authors used both "bifactor model" and "four-factors hierarchical model"). Using only one term may make the manuscript easier to read.
- To assess the fit of models, I suggest also considering the ratio of chi-squared to degrees of freedom (χ2 /df).
- p.15 – I suggest that the Authors also provide the full name of the instrument first (followed by acronymous). Also, to be consistent across instruments, the Authors should provide a brief description and scoring procedure for the instrument used to investigate "individual differences in personality traits and demographic characteristics".
- Tables – it would be helpful if the Authors could include the name of the study to which each table refers.
- The manuscript would benefit if the Authors could include a graphical representation of the selected model, including standardized loadings and residuals.

Experimental design

see Section 1

Validity of the findings

see Section 1

---

## Round 0.2 · accepted · Accept

You have addressed all of the reviewers' comments, and thus this manuscript is ready for publication

Note the comment from reviewer 1:

> Correct the name of the Italian author of HGSHS:A norms which is De Pascalis not Pascalis

·

Basic reporting

This ms has improved a lot. Now it is made clear that there are many different ways to understand and measure suggestibility and this method is just one of them. Correct the name of the Italian author of HGSHS:A norms which is De Pascalis not Pascalis

Experimental design

No comments

Validity of the findings

No commnets

Additional comments

No comments

·

Basic reporting

It looks like the authors have sufficiently responded to the review teams comments.

Experimental design

It looks like the authors have sufficiently responded to the review teams comments.

Validity of the findings

It looks like the authors have sufficiently responded to the review teams comments.

Additional comments

It looks like the authors have sufficiently responded to the review teams comments.

Reviewer 3 ·

Basic reporting

I am pleased with the revised manuscript on the self-report measure for assessing suggestibility. The revision addresses all of my concerns with the original manuscript, making it a valuable contribution to the literature on this topic.

Experimental design

All my requests have been met

Validity of the findings

All of my requests have been taken care of. In revising the manuscript, Authors have provided results backed up by extensive and sound analysis.